# CircuitProbe: Dissecting Spatiotemporal Visual Semantics with Circuit Tracing

## Abstract

The processing mechanisms underlying language and image understanding in large vision-language models (LVLMs) have been extensively studied. However, the internal reasoning mechanisms of LVLMs for *spatiotemporal understanding* remain poorly understood. In this work, we introduce a *systematic*, *circuit-based* framework designed to investigate how spatiotemporal visual semantics are represented and processed within these LVLMs. Specifically, our framework comprises three circuits: ① visual auditing circuit, ② semantic tracing circuit, and ③ attention flow circuit. Through the lens of these circuits, we discover that visual semantics are highly localized to specific object tokens–removing these tokens can degrade model performance by up to 92.6%. Furthermore, we identify that interpretable concepts of objects and actions emerge and become progressively refined in the middle-to-late layers of LVLMs. In contrary to the current works that solely focus on objects in one image, we reveal that the middle-to-late layers of LVLMs exhibit specialized functional localization for spatiotemporal semantics. Our findings offer significant mechanistic insights into spatiotemporal semantics analysis of LVLMs, laying a foundation for designing more robust and interpretable models.

## 1 Introduction

Large vision-language models (LVLMs) have emerged as powerful tools for understanding multimodal data by integrating video and linguistic information to produce text outputs (Liu et al., 2023b;a; Maaz et al., 2023; Zhang et al., 2024b). The predominant architecture for LVLMs integrates a pre-trained visual encoder with a pre-trained large language model (LLM) via a trainable adapter network (Li et al., 2022; Moon et al., 2023; Chen et al., 2023). This adapter module performs cross-modal feature alignment by converting the image encoder's visual semantics into continuous token embeddings (*i.e.*, soft prompts) that are compatible with the language model's input space (Merullo et al., 2023; Liu et al., 2024; Yu et al., 2025; Pan et al., 2025).

A systematic characterization of video semantics's impact on language modality is essential, as it governs: **(I)** the quality of vision-language alignment(Radford et al., 2021a; Zhang et al., 2024a); and **(II)** the design principles for high-performance LVLMs with robust reasoning capabilities (Pang et al., 2024; Woo et al., 2024; Park et al., 2024; Wang et al., 2025a). Unfortunately, most prior research focused on the interpretability of the visual encoder's embedding generation (Tong et al., 2024; Rajaram et al., 2024; Vilas et al., 2023), or its interaction with text tokens with individual image inputs (Palit et al., 2023; Hakimov & Schlangen, 2023), with limited focus on understanding the underlying mechanisms for spatiotemporal reasoning. The limitations of these methods are evident: ❶ they lack an in-depth exploration of how LLMs utilize visual information, and ❷ they fail to analyze how visual semantics interact with the discrete semantics within LLMs across the *spatiotemporal* dimension. In other words, the ways in which LLMs interact with spatiotemporal-rich visual data like videos remain largely unexplored, and demand deeper investigation (Yin et al., 2023).

Figure 1: *Our Tracing Circuit Framework*. We systematically analyze LVLMs by decomposing the circuit's information flow into three modules: ① visual auditing, ② semantic tracing, and ③ attention flow.

Existing video LVLMs typically leverage image-pretrained models and adapt them to video understanding via fine-tuning on video data. The mainstream methods achieve this mostly by designing projectors, such as Q-former (Li et al., 2023a), MLPs (Li et al., 2023b; Jin et al., 2024), or spatiotemporal pooling layers (Cheng et al., 2024) that can explicitly aggregate the spatiotemporal information contained in multiple video frames from a CLIP encoder (Radford et al., 2021b), or by continually pre-training the CLIP encoder on video data to better capture video information (Lin et al., 2023). However, such approaches inherently introduce biases because the structure and meaning of soft prompts derived from video are unclear, given that they do not map directly to discrete language tokens (Merullo et al., 2023). This raises a crucial question: how are these soft video representations understood and utilized during language model decoding, and do existing interpretability tools for language input generalize to this new modality?

Therefore, in this paper, we systematically evaluate the entire pipeline of LVLMs by dissecting it into three distinct *circuits*, meticulously tracing the information flow within each, as shown in Fig. 1. Specifically, **Circuit ①** aims to address whether visual semantics can be explicitly traced in both visual and semantic spaces. In this circuit, we examine how specific visual semantics are localized within visual tokens after projection, and assess changes in model performance when text injection is applied as an interference. Going further, **Circuit ②** delves into how visual semantics are processed at the neuron level within LLMs. Specifically, we utilize the language head to unembed hidden states into the explicit semantic space, and observe how knowledge evolves from shallow to deep layers. Lastly, **Circuit ③** seeks to understand how the model generates content given a specific visual context. We intervene in this reasoning process by blocking attention flow for specific layers or tokens, and then observe the corresponding changes in model performance. By carefully tracing the information flow within each circuit, we list key findings and principal contributions as follows:

❖ **Emergent Semantics.** We observe that video concept outputs from **Circuit ①**, after undergoing cross-modality attention alignment (**Circuit ③**), can be semantically traced by LLM **Circuit ②**. During LVLM inference, unlike existing image-trained LVLMs (Darcet et al., 2024), the concepts for video embedding are highly localized to the patch positions corresponding to their original location in each frame. To validate this, we ablate the object tokens and observe a significant performance degradation of 63.23% in the `LLaVA-NeXT` and 41.83% in `LLaVA-One-Vision` (`LLaVA-OV`) series models.

❖ **Knowledge Evolution.** During the forward pass in **Circuit ②**, empirical results reveal that visual soft prompts drive knowledge evolution, leading to the concurrent emergence of explainable object tokens and related temporal concepts (e.g., actions, position changes) in the mid to late layers. Specifically, a higher correspondence rate in 25∼30-th layers in `LLaVA-NeXT` and 20∼25-th layers in `LLaVA-OV`.

❖ **Functional Localization.** We demonstrate clear functional localization within the LLM backbone, as depicted in **Circuit** ③. Our findings indicate that the middle-to-late layers are most critical for interpreting object information from video frames. Through ablation studies, we measured the performance degradation of 29.5% in accuracy when masking object-related tokens specifically from layers 15 to layer 25.

## 2 BACKGROUND

In this section, we describe the methodology for processing images and queries through LVLMs and present the experimental details related to our ablation studies.

### 2.1 NOTATION

**Video Pre-processing.** The input frame $V$ undergoes pre-processing, where it is first cropped into a square and resized accordingly. The frame is then divided into a total of $N$ image patches. The CLIP ViT-L/14 image encoder, $f_C$, processes these patches to generate initial features. An adapter network then maps these features to the language model's input space, resulting in a set of $N$ visual tokens $E_V = \{e_1, \ldots, e_N\} \in \mathbb{R}^{N \times d_t}$, where $d_t$ is the dimensionality of the language model's input embeddings.

**Combined Input.** For the text input, given a tokenized prompt sequence $Q = (q_1, \ldots, q_M)$, the embedding layer of the language model LM maps these tokens into embeddings: $E_Q = \text{LM}(Q) \in \mathbb{R}^{M \times d_t}$. The final input to the language model is formed by concatenating the adapted image embeddings and the text embeddings: $X = [E_V; E_Q] \in \mathbb{R}^{(N+M) \times d_t}$ which is subsequently fed into the language model.

**Dataset Curation.** We curate the challenging subset of samples from the STAR benchmark (Wu et al.) to isolate tasks requiring genuine video understanding. First, we select key frames with clear actions and retain only questions where video information is indispensable, filtering out those solvable by static images or common sense. We then keep the sample only if LVLM correctly answers with the original frame but fails when the critical object is masked. This process evaluates the model's visual reasoning capabilities, independent of contextual biases. Sample frames are provided in Appendix A.

## 3 ANALYZING VISUAL SEMANTICS VIA CIRCUITS

We propose the hypothesis-driven methodology to dissect the internal computational circuits of LVLMs. Our approach surgically manipulates tokens and tracks their transformation across layers to explicitly test the predictions of our theoretical framework (detailed in Appendix B).

### 3.1 CIRCUIT ①: VISUAL INFORMATION AUDITING

**Methodology.** Circuit ① focuses on assessing the impact of specific visual information by manipulating video embeddings. We define $E_V$ as the set of visual tokens extracted from a video. we then identify the subset $E_O \subseteq E_V$ hypothesized to contain information about the particular object of interest. The remaining tokens, $E_C = E_V \setminus E_O$, are considered contextual. As illustrated in Fig. 2, our approach modifies the video embedding by replacing these hypothesized object tokens $E_O$ with *uninformative embeddings*. Specifically, these uninformative embeddings are computed as the mean embedding across all visual tokens derived from 10,000 images randomly sampled from the ImageNet (Deng et al., 2009) validation dataset. This replacement strategy allows us to precisely control the presence of object-specific visual information within the video embeddings without introducing extraneous noise. To evaluate the impact of this manipulation, we design two distinct question formats:

- **Open-ended questions:** We formulate specific questions targeting both objects and actions depicted in the video, such as "Which object did the person throw in the video?" Crucially, these questions are designed to be answered solely based on the provided video key frames. To facilitate a controlled comparison, we

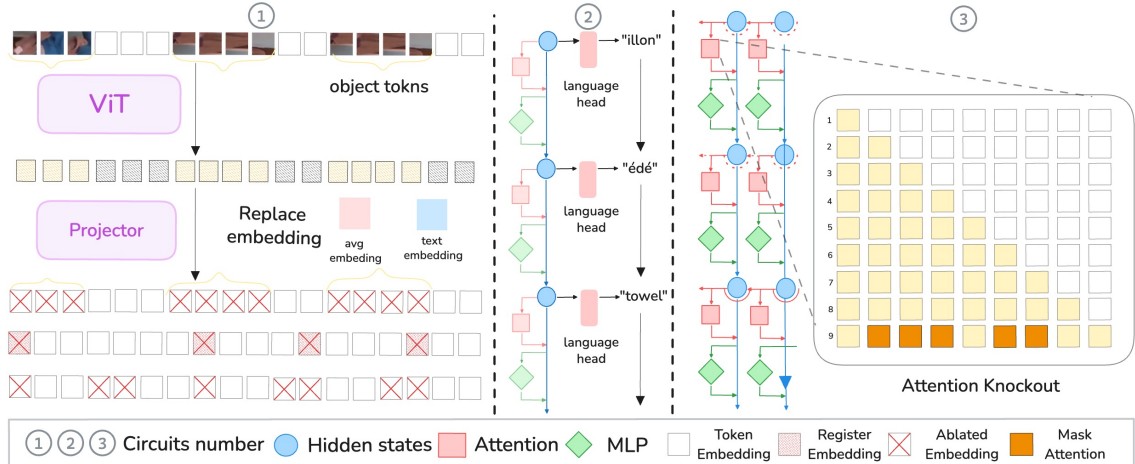

Figure 2: *Overview of the circuits experiments*. Our methodology comprises three key interventions: ① Ablating specific visual token subsets; ② tracing the semantic evolution of tokens across layers using logit lens; and ③ masking attention pathways to analyze information flow within the LVLM.

pre-fill the model's response with a prompt "The object is" and then analyze the next generated token before and after the visual token ablation.

- **Close-ended questions:** We prompt the model with binary questions, for instance, "Is there a/an [object] in this video?" We then assess the impact of ablation by comparing the model's next token prediction. A change from "Yes" to "No" after the ablation of tokens in $E_O$ serves as an indicator that the ablated tokens were crucial for the model's identification of the object.

We choose the subset of tokens to ablate in four ways: **(1) Random Tokens:** as a baseline, we ablate $n$ tokens in the video embedding. $E_R \subset E_V$ **(2) Register Tokens:** Following Darcet et al. (2024), we identify register tokens that encode global image features. Specifically, we select tokens whose L2 norms are more than two standard deviations above the mean norm within each key frame. **(3) Object Tokens:** We select a set of tokens $E_O$ corresponding to those image patches fully contained within the object's bounding box. **(4) Object Tokens with buffer:** We expand the object token set $E_O$ to include spatially adjacent tokens. We experiment with two buffer sizes: 1-Buffer, which includes immediately neighboring tokens, and 2-Buffer, which includes all tokens within a two-position radius of any object token.

*Observation ❶.* **The performance drop caused by removing object-specific tokens was substantially more pronounced than that observed when ablating control tokens.** As shown in Table 1, selectively replacing tokens from object regions with uninformative embeddings significantly degrades the model's question-answering performance. For instance, on LLaVA-NeXT-V, ablating approximately 573 object tokens caused a catastrophic 92.6% performance drop on open-ended questions. On the other hand, removing a far greater number of random tokens (900) resulted in a minimal drop of only 10.7%. This finding strongly suggests that crucial semantic information is embedded within the video tokens and is spatially localized to object-relevant regions of the frame. To mitigate potential model-specific bias and validate the robustness of our findings, we replicated these experiments on both LLaVA-NeXT and LLaVA-OV, whose variants were trained on only images (-I) or on videos and images (-V), observing consistent results across all models.

*Observation ❷.* **Text injection of semantic concepts led to significant improvement in the model's question-answering performance.** Motivated by our ablation findings, we investigated whether explicitly providing object concepts as text could enhance performance. We performed the text injection experiment (Fig. 3), where we replaced the object's visual tokens with the embedding of its textual label (e.g., "towel"). The results were striking. In our open-ended QA setting, models that had previously failed due to object

Table 1: Accuracy (%) from visual token ablation on question-answering performance across various models. The ↓ symbol indicates the magnitude of this performance drop. For object-based ablation, 'm/n' represents the average number of ablated tokens in the LLaVA-NeXT and LLaVA-OV models, respectively. Ablating object tokens causes a substantially greater performance drop compared to ablating an equivalent number of control tokens, demonstrating that crucial semantic information is spatially localized.

| Ablation Type | Tokens Number | LLaVA-NeXT-I | | LLaVA-NeXT-V | | LLaVA-OV-I | | LLaVA-OV-V | |
|---|---|---|---|---|---|---|---|---|---|
| | | Open | Close | Open | Close | Open | Close | Open | Close |
| *Control Groups (Low Tokens)* | | | | | | | | | |
| Baseline | 0 | $100.0_{\downarrow 0}$ | $100.0_{\downarrow 0}$ | $100.0_{\downarrow 0}$ | $100.0_{\downarrow 0}$ | $100.0_{\downarrow 0}$ | $100.0_{\downarrow 0}$ | $100.0_{\downarrow 0}$ | $100.0_{\downarrow 0}$ |
| Register | 13 | $96.6_{\downarrow 3.4}$ | $98.3_{\downarrow 1.7}$ | $98.0_{\downarrow 2.0}$ | $98.2_{\downarrow 1.8}$ | $84.7_{\downarrow 15.3}$ | $98.2_{\downarrow 1.8}$ | $90.3_{\downarrow 9.7}$ | $96.9_{\downarrow 3.1}$ |
| *Object-based Ablation* | | | | | | | | | |
| | 304/404 | $18.1_{\downarrow 81.9}$ | $45.9_{\downarrow 54.1}$ | $17.3_{\downarrow 82.7}$ | $62.3_{\downarrow 37.7}$ | $40.5_{\downarrow 59.5}$ | $67.9_{\downarrow 32.1}$ | $60.8_{\downarrow 39.2}$ | $78.8_{\downarrow 21.2}$ |
| Object | 413/539 | $10.0_{\downarrow 90.0}$ | $34.5_{\downarrow 65.5}$ | $12.5_{\downarrow 87.5}$ | $49.3_{\downarrow 50.7}$ | $33.6_{\downarrow 66.4}$ | $53.3_{\downarrow 46.7}$ | $52.6_{\downarrow 47.4}$ | $67.1_{\downarrow 32.9}$ |
| | 573/694 | $9.3_{\downarrow 90.7}$ | $27.8_{\downarrow 72.2}$ | $7.4_{\downarrow 92.6}$ | $44.9_{\downarrow 55.1}$ | $30.9_{\downarrow 69.1}$ | $45.9_{\downarrow 54.1}$ | $51.7_{\downarrow 48.3}$ | $63.0_{\downarrow 37.0}$ |
| *Control Groups (High Tokens)* | | | | | | | | | |
| | 100 | $97.7_{\downarrow 2.3}$ | $99.1_{\downarrow 0.9}$ | $97.7_{\downarrow 2.3}$ | $98.4_{\downarrow 1.6}$ | $85.2_{\downarrow 14.8}$ | $98.0_{\downarrow 2.0}$ | $90.5_{\downarrow 9.5}$ | $97.1_{\downarrow 2.9}$ |
| | 350 | $94.0_{\downarrow 6.0}$ | $98.8_{\downarrow 1.2}$ | $95.5_{\downarrow 4.5}$ | $98.7_{\downarrow 1.3}$ | $84.7_{\downarrow 15.3}$ | $97.7_{\downarrow 2.3}$ | $89.6_{\downarrow 10.4}$ | $96.5_{\downarrow 3.5}$ |
| Random | 500 | $94.0_{\downarrow 6.0}$ | $98.6_{\downarrow 1.4}$ | $93.3_{\downarrow 6.7}$ | $98.7_{\downarrow 1.3}$ | $84.9_{\downarrow 15.1}$ | $97.7_{\downarrow 2.3}$ | $89.4_{\downarrow 10.6}$ | $96.5_{\downarrow 3.5}$ |
| | 700 | $90.6_{\downarrow 9.4}$ | $98.3_{\downarrow 1.7}$ | $91.4_{\downarrow 8.6}$ | $98.4_{\downarrow 1.6}$ | $85.4_{\downarrow 14.6}$ | $97.7_{\downarrow 2.3}$ | $88.6_{\downarrow 11.4}$ | $96.7_{\downarrow 3.3}$ |
| | 900 | $89.4_{\downarrow 10.6}$ | $98.0_{\downarrow 2.0}$ | $89.3_{\downarrow 10.7}$ | $98.4_{\downarrow 1.6}$ | $85.3_{\downarrow 14.7}$ | $97.2_{\downarrow 2.8}$ | $90.4_{\downarrow 9.6}$ | $97.1_{\downarrow 2.9}$ |

Figure 3: ***Text injection experiment***. The result shows the performance change when visual object tokens are replaced by their corresponding embedded textual labels. Error injection indicates wrong concept injection.

ablation (e.g., 17.3% accuracy) achieved an accuracy of 82.9% after text injection. This demonstrates not only the complete reversal of the performance degradation but significant overall improvement. This finding yields two critical insights: First, models can effectively ground its reasoning in explicit textual semantics. Second, the symbolic text label provides a cleaner, more powerful semantic signal than the corresponding visual tokens. This confirms that the model's performance is fundamentally dependent on object-level conceptual information. We conducted supplementary experiments to verify that our conclusions generalize to larger-scale model size in Appendix C.

> **Takeaway ❶.** Object-specific semantics are spatially localized within visual tokens. Replacing these visual tokens with their symbolic text labels significantly boosts accuracy, indicating that semantic-rich conceptual inputs more effectively drive the model's reasoning than by the raw, and potentially noisy, visual features themselves.

## 3.2 Circuit ②: Semantic Tracing

Our goal is to understand where and how the model translates raw visual inputs into abstract textual meanings. To achieve this, we employ the logit lens approach (Nostalgebraist, 2020), which allows us to probe the intermediate representations at each layer. Specifically, for any given video token $e_i$ at layer $l$, we decode its

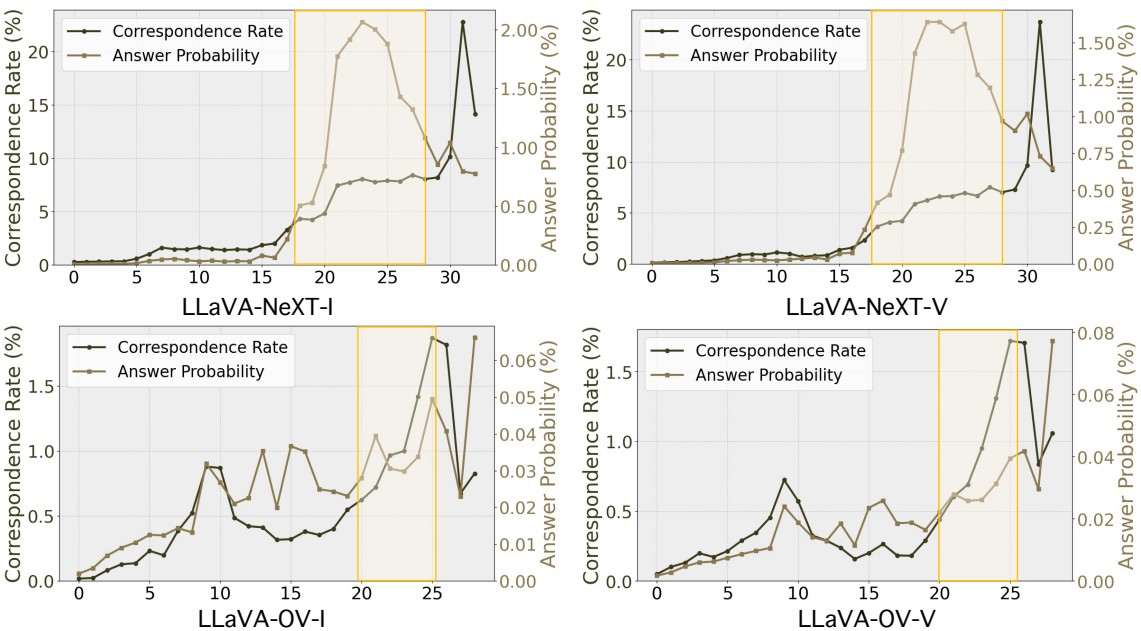

Figure 4: Quantitative analysis of semantic tracing. Both metrics show a sharp increase in the mid-to-late layers, indicating that abstract semantic concepts are consolidated deep within the network.

hidden state $h_i^{(l)}$ by projecting it through the model's language head, $W_{LM}$, to observe the vocabulary distribution it implies. The top-ranked word reveals the model's "interpretation" of that visual patch at that stage of processing. Using this technique, we perform both quantitative and qualitative analyses to investigate the emergence of object-level concepts and the model's ability to capture temporal dynamics.

**Metric Definition.** We propose two metrics to quantitatively track the emergence of semantic concepts within embeddings across layers. For a given object represented by N visual tokens, let $h_i^{(l)}$ denote the hidden state of the $i$-th token at layer $l$. Our metrics evaluate how accurately this hidden state can be decoded into the correct textual label $w_{correct}$ from the vocabulary $W$. Our analysis focuses exclusively on the internal representations of visual tokens within the LLM backbone during the prefill stage.

- *Correspondence Rate $C_R^{(l)}$* is a hard metric that measures the fraction of all visual tokens at a given layer $l$ whose hidden states decode to the correct semantic label of the primary object.

$$C_R^{(l)} = \frac{1}{N} \sum_{i=1}^{N} \mathbb{I}\left( \underset{w \in W}{\arg\max} \left( \text{softmax}(W_{LM} h_i^{(l)}) \right)_w = w_{\text{correct}} \right). \tag{1}$$

- *Answer Probability $A_P^{(l)}$* offers a softer measure of confidence by computing the average logit probability of the correct object token across all visual tokens at layer $l$.

$$A_P^{(l)} = \frac{1}{N} \sum_{i=1}^{N} \frac{\exp\left( (W_{LM} h_i^{(l)}) w_{\text{correct}} \right)}{\sum_{w' \in W} \exp\left( (W_{LM} h_i^{(l)}) w' \right)}. \tag{2}$$

*Observation ❸. $C_R^{(l)}$ and the $A_P^{(l)}$ increase sharply in the mid-to-late layers.* As shown in Fig. 4, abrupt rise in the mid-to-late layers suggests that abstract semantic representations are largely absent in early layers

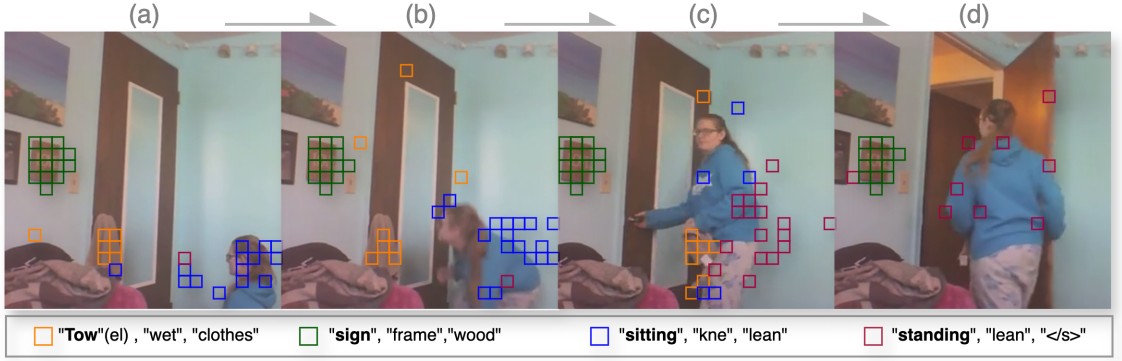

Figure 5: Qualitative example of semantic tracing illustrating how the model captures *temporal dynamics*. The sequence from (a) to (d) shows the evolution of the top-3 most frequent word groups for the single token position. The predictions shift from semantics related to an initial state (e.g., *sitting*) to those reflecting the completed action (e.g., *standing*), highlighting the model's ability to track actions over time.

and are instead progressively formed and refined as features propagate through the network. Late-layer visual tokens demonstrate the strong spatial correspondence, where each token's embedding aligns with the semantic identity of the object in its original patch location.

*Observation* ❹. **Visual tokens in late layers locally encode object semantics.** We find that the activation vector for an individual visual token in the final layers aligns closely with the text embedding of the object contained within its corresponding image patch. As illustrated in Fig. 5, a visual token from a patch containing "towel" develops representation with high semantic similarity to the text embedding of the word "towel". This demonstrates that the model consolidates abstract object representations at the individual token level.

*Observation* ❺. **LVLMs exhibits strong capabilities in capturing temporal dynamics for action recognition.** As shown in Fig. 5, for a sequence where a person stands up, we observe that tokens semantically related to "sitting" emerge in the initial frames. These tokens then diminish and are replaced by tokens corresponding to "standing" in the later frames. This progression demonstrates the model's capacity to track the evolution of actions over time and make predictions based on the most salient temporal cues.

> **Takeaway ❷.** By tracing representations layer-by-layer, we pinpoint the emergence of semantic understanding to the model's *mid-to-late* layers. Abstract concepts, absent in early layers, sharply consolidate deep in the network, where individual visual tokens evolve to match the symbolic meaning of their patch content. This dynamic, token-level consolidation enables the model to not only identify objects but also track the evolution of actions over time (more interactive samples in D).

### 3.3 CIRCUIT ③: ATTENTION FLOW

Our prior experiments demonstrated the presence of object-specific information in a particular region of the video embedding; however, it remained unclear whether the contextual information within these embeddings inherently encoded temporal details processable by an LLM decoder. In this section, we investigate whether LLMs can leverage localized object information within video embeddings for temporal reasoning. Specifically, we explore if this can occur without reliance on the broader contextual cues. This investigation aims to disambiguate the extent to which LLMs can derive temporal understanding from isolated, object-centric features. We conduct all experiments on the `LLaVA-NeXT` series model.

**Methodology.** To precisely control the information flow during the LLM's prediction of the first token, we introduce a targeted attention masking strategy. This mask is applied to hypothesize and restrict attention between different token and layer subsets. We partition the LLM's layers into five distinct windows: Early layers, Early-to-middle layers, Middle layers, Middle-to-late layers, and Late layers. Specifically, we define

the mask for all heads at each window. For each of these windows, we quantify the impact of blocking attention on two key metrics: the decrease in accuracy and the prediction probability degradation for the correct token. This analysis is conducted under two distinct masking scenarios:

- **O**bject-centric masking: Attention is blocked from object tokens $E_O$ and their buffer to the final token.
- **C**ontextual masking: Attention is blocked from all non-object tokens $E_C$ to the final token.

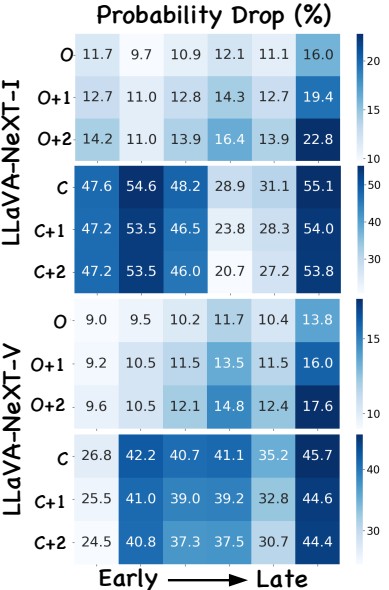

| Model | Condition (From*) | Early | Early-Mid | Mid | Mid-Late | Late | All Layers |
|---|---|---|---|---|---|---|---|
| *Object-centric masking* | | | | | | | |
| | O | 0.19 | 0.20 | 0.24 | **0.25** | 0.23 | 0.29 |
| LLaVA-NeXT-I | O+1 | 0.20 | 0.23 | 0.25 | **0.28** | 0.27 | 0.35 |
| | O+2 | 0.23 | 0.23 | 0.26 | **0.28** | 0.26 | 0.39 |
| | O | 0.25 | 0.26 | 0.27 | **0.30** | 0.27 | 0.32 |
| LLaVA-NeXT-V | O+1 | 0.25 | 0.26 | 0.29 | **0.33** | 0.29 | 0.38 |
| | O+2 | 0.25 | 0.26 | 0.31 | **0.33** | 0.32 | 0.42 |
| *Contextual masking* | | | | | | | |
| | C | 0.82 | **0.99** | 0.89 | 0.66 | 0.44 | 0.99 |
| LLaVA-NeXT-I | C+1 | 0.83 | **0.97** | 0.86 | 0.59 | 0.39 | 0.97 |
| | C+2 | 0.83 | **0.97** | 0.86 | 0.54 | 0.39 | 0.97 |
| | C | 0.49 | **0.91** | 0.89 | 0.92 | 0.62 | 0.97 |
| LLaVA-NeXT-V | C+1 | 0.47 | **0.90** | 0.86 | 0.88 | 0.56 | 0.94 |
| | C+2 | 0.46 | **0.90** | 0.83 | 0.87 | 0.52 | 0.92 |

Table 2: **Accuracy drop** for `LLaVA-NeXT` models across various layer windows after attention knockout. Masking conditions: **O**, object tokens; **O+1/2**, object with 1/2 buffer; **C**, all tokens except the object; **C+1/2**, all tokens except the object with 1/2 buffer.

Figure 6: **Answer probability drop** under attention knockout across various layers.

**Analysis.** This structured approach allows us to investigate the specific contributions of object-related versus general contextual information at different depths of the LLM.

*Observation ❻*. **Masking contextual information in LLMs incurs greater performance degradation than masking object-centric regions when generating the first token.** This finding suggests that the models rely on contextual cues for reasoning, rather than simply retrieving the object to formulate an answer(As shown in Table 2 and Fig. 6). Our results further indicate that contextual and fine-grained information are processed at different network depths when LLM generated the first token.

*Observation ❼*. **The performance drop from masking contextual information is the most pronounced in the early-to-mid layers, whereas masking fine-grained details primarily impacts the mid-to-late layers.** This suggests a two-stage reasoning process: models first integrate contextual information in earlier layers before shifting focus to fine-grained details in later layers to generate the final output. Interestingly, the layers that are most sensitive to masking differ between image-trained and video-trained models. For image-trained models, the performance degradation is concentrated in the early-to-mid layers, while for video-trained models, the impact is centered on the mid-layers.

> **Takeaway ❸.** LVLM employs "two-stage" reasoning process when generating the first token. It first grounds its linguistic state by integrating broad visual context in the *early-to-mid layers*. Subsequently, it refines its prediction by focusing on localized object details in the *mid-to-late layers*. This demonstrates a context-first, detail-later strategy for translating visual dynamics into language.

## 4 RELATED WORK

### 4.1 INTERPRETABILITY OF LARGE LANGUAGE MODELS

Recent research has increasingly focused on LLM interpretability, with three mainstream approaches emerging in this domain(Singh et al., 2024; Wang et al., 2025b). The first approach employs circuit analysis methods, which assume that only a subset of parameters within the model are crucial, thereby allowing the model to be simplified into a sparse circuit; representative works include Hanna et al. (2023), who constructed computational graphs to understand numerical comparison mechanisms, and Yao et al. (2024), who explored knowledge circuit variations across different scenarios such as knowledge editing and in-context learning. The second research approach, known as causal tracing, focuses on tracking causal pathways during model data processing to analyze modules that contribute significantly to outputs. This line of research, pioneered by ROME (Meng et al., 2022a), has led to methods for precisely editing factual knowledge within models and has been extended to various architectures and tasks (Meng et al., 2022b; Sharma et al., 2024; Geva et al., 2023; Zhao et al., 2023; Yu & Ananiadou, 2024). The third approach utilizes unembedding space projection, where internal model representations are projected onto token spaces to gain interpretability insights. Geva et al. (2021) demonstrated this concept for feed-forward networks, while Dar et al. (2023) extended it to other model parameters, showing that both training and fine-tuning parameters can be interpreted within embedding spaces. While these methods have considerably enhanced our understanding of language model internals, they focus predominantly on textual processing. Multimodal interpretability, particularly the analysis of visual-textual integration remains an under-explored frontier.

### 4.2 INTERPRETABILITY OF VIDEO LANGUAGE MODELS

Existing work on visual model interpretability primarily focuses on two key areas: visual encoder embedding generation and the interaction between visual inputs and textual tokens. Research on visual encoder embedding generation has explored the integration of pre-trained language models with visual processing, with works investigating how frozen transformer blocks can process visual tokens, how attention misalignment leads to hallucinations, and how visual and textual embedding spaces can be effectively bridged (Pang et al., 2024; Woo et al., 2024; Park et al., 2024). Studies examining visual-textual interactions have focused on understanding fundamental visual processing capabilities in multimodal systems, including work on identifying systematic shortcomings in vision-language models (Tong et al., 2024; Verma et al., 2024), extracting computational subgraphs for visual concept recognition (Rajaram et al., 2024), and reverse-engineering Vision Transformers to understand categorical representation building (Vilas et al., 2023). More recently, research has begun addressing video data processing in large models, with Li et al. (2024) identifying temporal reasoning bottlenecks in Video LLMs and Joseph et al. (2025) developing tools for accelerating visual mechanistic interpretability research. However, these studies have not yet analyzed how visual semantics interact with LLMs' internal discrete semantics across spatiotemporal dimensions, leaving a significant gap in understanding how LLMs process complex visual information over time.

## 5 CONCLUSION AND LIMITATION

In this paper, we introduced a circuit-based framework to provide a mechanistic interpretation of spatiotemporal reasoning in LVLMs, revealing a clear information processing pipeline. Our analysis shows that visual semantics are first spatially localized in specific object tokens. These tokens are then processed by the LLM, where abstract object and action concepts emerge and consolidate in the mid-to-late layers. Finally, the model exhibits functional localization through a two-stage reasoning process: it grounds its understanding in the broad context using early-to-mid layers before refining its answer with object-specific details in the mid-to-late layers. These findings offer a coherent explanation of LVLM reasoning, moving beyond black-box evaluations toward a more principled understanding. However, our work remains observational, as we have not yet leveraged these findings through interventional methods like "circuit surgery" to causally probe the reasoning process or enhance model robustness against failures like hallucinations. Future work will focus on leveraging the findings to create more robust and interpretable LVLMs.

USAGE OF LARGE LANGUAGE MODELS

We declare that LLMs were used solely for language polishing purposes in this work. Specifically, after completing the initial draft entirely through human effort, we employed LLM assistance exclusively for grammatical refinement and improving the clarity of English expression to meet academic writing standards. All intellectual contributions, from conceptualization to initial manuscript preparation, were performed by the human authors. The use of LLM was limited to post-writing language enhancement, similar to traditional proofreading services, ensuring that non-native English speakers can present their research with appropriate linguistic quality while maintaining complete authorship and originality of the scientific content.

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

APPENDIX

## A    FRAMES VISUALIZATION

This section visualizes sample input keyframes from the videos to provide a clearer understanding of the data used in our experiments. Each row in Figure 7 represents a distinct video sequence fed into the LVLMs for analysis. These examples are representative of the scenarios in our dataset, encompassing a variety of everyday actions, objects, and environments. By visualizing the raw inputs, we aim to illustrate the visual complexities, such as changes in viewpoint, object scale, and partial occlusions, that the model must handle to perform accurate semantic reasoning.

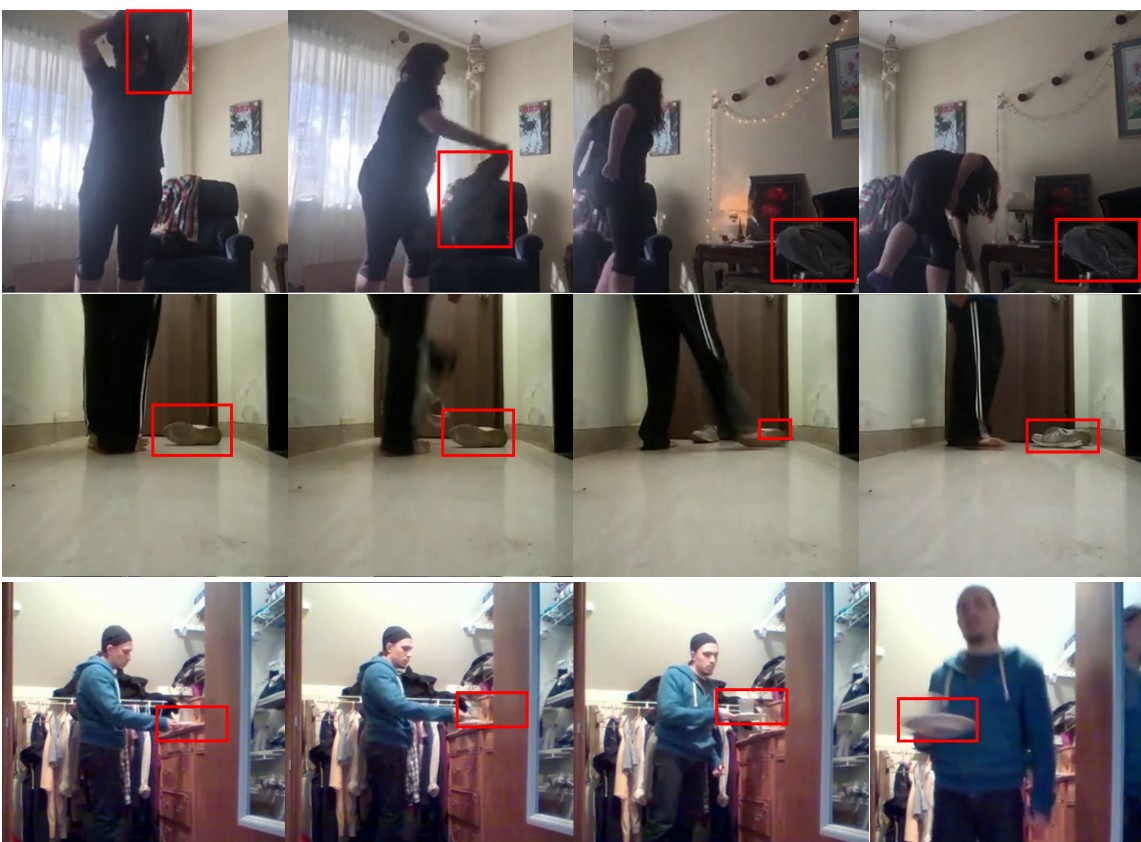

Figure 7: Visualization of Input Keyframes. Each row displays a sequence of frames provided to the model as input for a specific video. The red bounding boxes highlight the ground-truth object pertinent to the task's question (e.g., the object being picked up, kicked, or taken). It is important to note that these bounding boxes are included here for clarity and were not provided to the model during inference.

## B    THEORETICAL FRAMEWORK FOR VISUAL SEMANTIC CIRCUITS

In this section, we propose the theoretical framework built on three core principles that we hypothesize govern the internal computations of LVLMs: Information Localization, Progressive Semantic Refinement, and a Two-Stage Reasoning flow. We model the LVLM as a probabilistic system to formalize these principles

into specific, falsifiable predictions. This framework provides a principled foundation for understanding and predicting the model's behavior, which we then validate through targeted experiments.

## B.1 PROBABILISTIC MODEL FORMULATION

Let $V$ be a video represented by a sequence of key frames, and $Q$ be a textual question. The LVLM, denoted by $\mathcal{M}$, aims to generate an answer $A$. The process begins by encoding the video $V$ into a set of $N$ visual tokens, $E_V = \{e_1, e_2, \ldots, e_N\}$. The question $Q$ is tokenized into $M$ text tokens, $E_Q = \{q_1, \ldots, q_M\}$. The model then computes the probability of an answer $A$:

$$P(A|V, Q) = \mathcal{M}(E_V, E_Q). \tag{3}$$

Central to our investigation is the hypothesis that the set of visual tokens $E_V$ can be partitioned into the subset $E_O$ containing primary information about the specific object $o$, and the complementary subset $E_C$ containing contextual information, such that $E_V = E_O \cup E_C$ and $E_O \cap E_C = \emptyset$.

## B.2 PRINCIPLE OF INFORMATION LOCALIZATION

We begin with the hypothesis that to answer a specific question, the LVLM does not treat all visual tokens equally. Instead, we propose the *Principle of Information Localization*: task-critical information is spatially concentrated in the subset of tokens corresponding to the object of interest. Let this subset be $E_O$, with the remainder being contextual tokens $E_C$.

This principle leads to a direct, testable prediction. The informational value of a token set can be quantified by the degradation in model performance upon its ablation. We model this degradation as the KL divergence between the original and ablated posterior distributions:

$$\mathcal{L}_{\mathrm{drop}}(E_S) = D_{KL}\left(P(A|E_V, E_Q) \parallel P(A|E_V \setminus E_S, E_Q)\right). \tag{4}$$

Our principle predicts that the information is concentrated in $E_O$. Formally, if we ablate the object tokens $E_O$, the resulting information loss should be significantly greater than ablating any other random subset of tokens $E_R$ of the same size. This leads to the following inequality, which we aim to verify experimentally:

$$\mathbb{E}_{E_R \subset E_V, |E_R|=|E_O|}\left[\mathcal{L}_{\mathrm{drop}}(E_R)\right] \ll \mathcal{L}_{\mathrm{drop}}(E_O). \tag{5}$$

Furthermore, we hypothesize that the model internally reasons over abstract concepts. This predicts that injecting a clean, symbolic representation of the object, $e_{w_{\mathrm{correct}}}$, should be even more effective than the noisy visual tokens $E_O$. This can be formalized as:

$$P(A^*|e_{w_{\mathrm{correct}}}, E_C, E_Q) > P(A^*|E_O, E_C, E_Q). \tag{6}$$

The ablation and injection experiments presented in Table 1 were designed to test these formal predictions.

## B.3 HYPOTHESIS OF PROGRESSIVE SEMANTIC REFINEMENT

We hypothesize that visual information is not processed into its final semantic form in a single step. Instead, we propose the model of *Progressive Semantic Refinement*, where hidden states associated with visual tokens transition from encoding low-level perceptual features in early layers to abstract, language-aligned concepts in later layers.

Let $h_i^{(l)}$ be the hidden state for a token $i$ at layer $l$. Let $\mathcal{S}(w_{\mathrm{correct}})$ be the semantic space associated with the correct object concept, represented by its text embedding $e_{w_{\mathrm{correct}}}$. Our hypothesis predicts that for an object token $i \in E_O$, its representation $h_i^{(l)}$ will become progressively more aligned with this semantic space as it passes through the network. We can formalize this predicted monotonic increase in alignment for layers $l$ beyond a critical depth $l_{\mathrm{crit}}$ using a similarity metric:

$$\mathrm{sim}(h_i^{(l)}, e_{w_{\mathrm{correct}}}) \quad \text{is a monotonically increasing function of } l \text{ for } l > l_{\mathrm{crit}}. \tag{7}$$

To test this prediction, we employ the logit lens technique, which projects intermediate hidden states into the vocabulary space. We measure the Correspondence Rate ($C_R^{(l)}$) and Answer Probability ($A_P^{(l)}$) to track this alignment across layers. The experimental results in Figure 4 are used to validate the existence and location of the predicted critical layer depth $l_{\text{crit}}$.

### B.4 THE TWO-STAGE REASONING HYPOTHESIS

Building on the previous principles, we hypothesize that the model's reasoning is not monolithic but follows an efficient, cognitively plausible two-stage process.

1. **Stage 1 (Contextual Grounding):** In the early layers ($\mathcal{L}_{\text{early}}$), the model first processes contextual tokens ($E_C$) to establish a general understanding of the scene and the query.

2. **Stage 2 (Focal-Point Refinement):** In the late layers ($\mathcal{L}_{\text{late}}$), after the context is established, the model focuses its attention on the specific object tokens ($E_O$) to extract fine-grained details necessary for a precise answer.

This hypothesis can be formalized by considering the sensitivity of the final prediction to attention weights at different layers. Let $\nabla_{\alpha_S^{(l)}} \log P(a_1^*)$ be the gradient of the log-probability of the correct answer with respect to the attention weights from a set of tokens $S$ at layer $l$. Our two-stage hypothesis predicts a shift in sensitivity:

$$\sum_{l \in \mathcal{L}_{\text{early}}} \left\| \nabla_{\alpha_C^{(l)}} \log P(a_1^*) \right\| > \sum_{l \in \mathcal{L}_{\text{early}}} \left\| \nabla_{\alpha_O^{(l)}} \log P(a_1^*) \right\|, \tag{8}$$

$$\sum_{l \in \mathcal{L}_{\text{late}}} \left\| \nabla_{\alpha_O^{(l)}} \log P(a_1^*) \right\| > \sum_{l \in \mathcal{L}_{\text{late}}} \left\| \nabla_{\alpha_C^{(l)}} \log P(a_1^*) \right\|. \tag{9}$$

These inequalities formalize the "context-first, detail-later" strategy as a testable prediction. We designed the attention masking experiments in Table 2 to directly probe these sensitivities and validate our two-stage reasoning hypothesis.

## C SCALING EXPERIMENTS

We conducted supplementary experiments to verify that our conclusions generalize to larger-scale models. We replicated our core analyses on the LLaVA-NeXT-34B model variants, with results that closely mirror those presented in the main body of the paper. The visual token ablation study on the 34B models reaffirms the principle of spatial localization for semantic information. Ablating object-specific tokens incurs significantly more substantial performance degradation than removing larger quantity of random tokens (in Table 3).

Furthermore, our semantic tracing analysis on the 34B architecture, depicted in Figure 8, reveals a conceptual emergence pattern consistent with our earlier observations. Both the Correspondence Rate and Answer Probability remain negligible through the initial layers before exhibiting a sharp, concurrent rise beginning around layer 40. This trend indicates that abstract, language-aligned concepts are consolidated in the deeper layers of the network, irrespective of model scale. These scaling experiments provide robust evidence that the mechanisms of semantic localization and late-stage conceptual formation are fundamental properties of the tested LVLM architectures.

| Ablation Type | Tokens Number | LLaVA-NeXT-34B-I | | LLaVA-NeXT-34B-V | |
|---|---|---|---|---|---|
| | | Open | Close | Open | Close |
| *Control Groups (Low Tokens)* | | | | | |
| Baseline | 0 | $100.0_{\downarrow 0}$ | $100.0_{\downarrow 0}$ | $100.0_{\downarrow 0}$ | $100.0_{\downarrow 0}$ |
| Register | 13 | $55.3_{\downarrow 44.68}$ | $96.9_{\downarrow 3.14}$ | $55.3_{\downarrow 44.68}$ | $1.8_{\downarrow 98.2}$ |
| *Object-based Ablation* | | | | | |
| | 304 | $9.3_{\downarrow 90.75}$ | $29.8_{\downarrow 70.16}$ | $14.6_{\downarrow 85.37}$ | $11.7_{\downarrow 88.35}$ |
| Object | 413 | $5.8_{\downarrow 94.24}$ | $21.1_{\downarrow 78.88}$ | $10.7_{\downarrow 89.32}$ | $11.3_{\downarrow 88.72}$ |
| | 573 | $5.8_{\downarrow 94.24}$ | $18.0_{\downarrow 82.02}$ | $10.3_{\downarrow 89.74}$ | $11.8_{\downarrow 88.21}$ |
| *Control Groups (High Tokens)* | | | | | |
| | 100 | $54.3_{\downarrow 45.72}$ | $96.9_{\downarrow 3.14}$ | $93.1_{\downarrow 6.92}$ | $29.7_{\downarrow 70.26}$ |
| Random | 350 | $55.5_{\downarrow 44.5}$ | $96.5_{\downarrow 3.49}$ | $88.5_{\downarrow 11.54}$ | $31.0_{\downarrow 68.97}$ |
| | 500 | $54.5_{\downarrow 45.55}$ | $96.3_{\downarrow 3.66}$ | $83.1_{\downarrow 16.92}$ | $30.8_{\downarrow 69.23}$ |
| | 900 | $59.9_{\downarrow 40.14}$ | $96.8_{\downarrow 3.17}$ | $78.3_{\downarrow 21.67}$ | $29.7_{\downarrow 70.33}$ |

Table 3: Accuracy (%) from visual token ablation on question-answering performance across 34B models. The ↓ symbol indicates the magnitude of this performance drop.

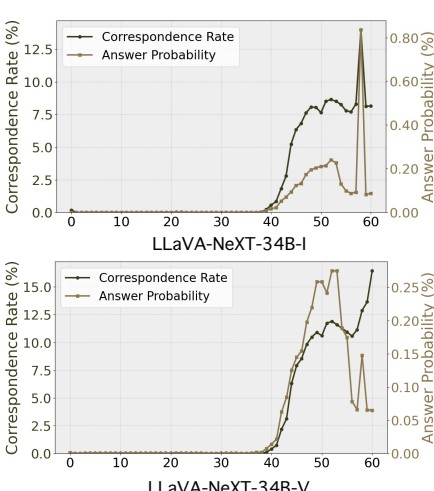

Figure 8: Quantitative analysis of semantic tracing on 34B model size.

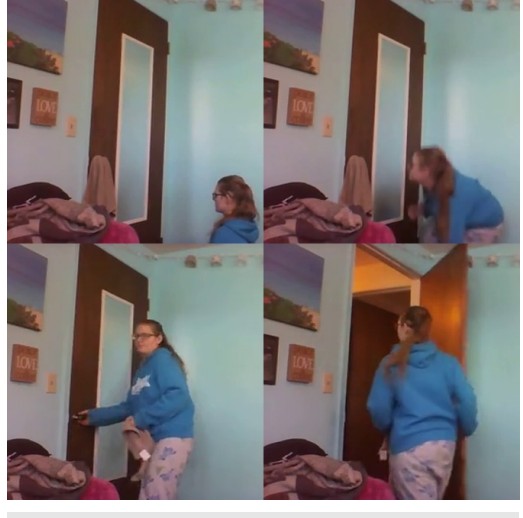

*Prompt*
USER: The input consists of a sequence of key frames from a video. Question: Which object was taken by the person?
ASSISTANT: The object is the

*Answer*: Towel

*Response:* Towel

| Token/Layer | Layer 1 | Layer 2 | Layer 3 | Layer 4 | Layer 5 |
|---|---|---|---|---|---|
|  | Institution | Архив | sierp | sierp | sierp |
| US | ona | Архив | pert | tap | pert |
| ER | Session | пута | oded | 庄 | pel |
| : | Portail | пута |  |  | Sav |
| | | пута | Carter | � | � |
| <IMG001> | sak | sak | sak | sak | sak |
| <IMG002> | gresql | gresql | gresql | gresql | gresql |
| <IMG003> | olas | olas | olas | olas | olas |
| <IMG004> | << | << | << | << | << |
| <IMG005> | gresql | gresql | gresql | gresql | gresql |
| <IMG006> | sak | sak | sak | sak | sak |
| <IMG007> | yter | yter | yter | yter | yter |
| <IMG008> | nahm | nahm | nahm | nahm | nahm |
| <IMG009> | loop | loop | loop | loop | sak |
| <IMG010> | loop | loop | loop | loop | loop |
| <IMG011> | gresql | gresql | gresql | gresql | gresql |
| <IMG012> | yter | yter | yter | yter | yter |
| <IMG013> | izi | izi | Emp | Emp | conf |
| <IMG014> | alle | alle | alle | alle | conf |
| <IMG015> | yter | yter | yter | yter | yter |
| <IMG016> | alle | alle | alle | arab | anno |
| <IMG017> | yter | yter | yter | yter | yter |

Figure 9: Qualitative example of the model correctly identifying an object. The user asks which object was taken by the person. The model correctly identifies the "Towel". The accompanying table shows the layer-by-layer semantic tracing for visual and text tokens.

# D QUALITATIVE EXAMPLES

We provide additional qualitative examples to visually illustrate the findings from our circuit-based analysis. These examples showcase the model's process of interpreting video frames to answer specific questions about objects and actions. Each figure includes the input video frames, the posed question, the model's response, and a table showing the semantic evolution of key tokens across different layers, as analyzed through our semantic tracing circuit (Circuit ②). More examples in the anonymous interactive demo website.

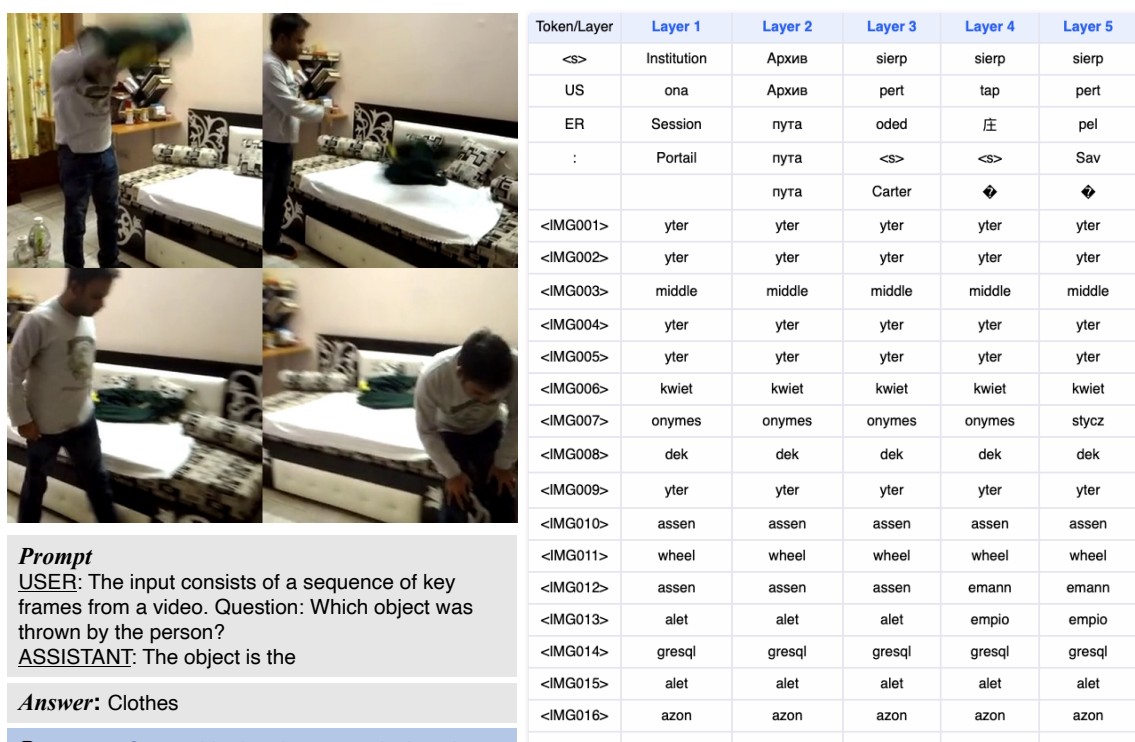

| Token/Layer | Layer 1 | Layer 2 | Layer 3 | Layer 4 | Layer 5 |
|---|---|---|---|---|---|
|  | Institution | Архив | sierp | sierp | sierp |
| US | ona | Архив | pert | tap | pert |
| ER | Session | пута | oded | 庄 | pel |
| : | Portail | пута |  |  | Sav |
| | | пута | Carter | � | � |
| <IMG001> | yter | yter | yter | yter | yter |
| <IMG002> | yter | yter | yter | yter | yter |
| <IMG003> | middle | middle | middle | middle | middle |
| <IMG004> | yter | yter | yter | yter | yter |
| <IMG005> | yter | yter | yter | yter | yter |
| <IMG006> | kwiet | kwiet | kwiet | kwiet | kwiet |
| <IMG007> | onymes | onymes | onymes | onymes | stycz |
| <IMG008> | dek | dek | dek | dek | dek |
| <IMG009> | yter | yter | yter | yter | yter |
| <IMG010> | assen | assen | assen | assen | assen |
| <IMG011> | wheel | wheel | wheel | wheel | wheel |
| <IMG012> | assen | assen | assen | emann | emann |
| <IMG013> | alet | alet | alet | empio | empio |
| <IMG014> | gresql | gresql | gresql | gresql | gresql |
| <IMG015> | alet | alet | alet | alet | alet |
| <IMG016> | azon | azon | azon | azon | azon |
| <IMG017> | fle | fle | fle | fle | wi |

**Prompt**
USER: The input consists of a sequence of key frames from a video. Question: Which object was thrown by the person?
ASSISTANT: The object is the

**Answer**: Clothes

*Response:* Green shirt that the person is throwing.

Figure 10: Qualitative example where the model is prompted to identify a thrown object. The model successfully responds that a "Green shirt" was thrown, correctly identifying both the object and its color. The table illustrates the semantic trace, showing how the model processes the visual information through its layers.

# E ETHICS STATEMENT

This research adheres to the ICLR Code of Ethics and aims to enhance the understanding of the internal spatiotemporal reasoning mechanisms within LVLMs through circuit-based analysis, in order to drive the development of more robust and interpretable models. The experiments are based entirely on public academic datasets (e.g., the STAR benchmark), and we acknowledge that these contain videos of human activities, which were used solely for their intended academic analytical purposes. To ensure research integrity, we explicitly state that LLMs were used only for language polishing after the manuscript was written, and that all core scientific contributions originate from the human authors.

## F    REPRODUCIBILITY STATEMENT

We are committed to ensuring the reproducibility of this research. All experiments are based on publicly available models (the `LLaVA-NeXT` and `LLaVA-OV` series) and datasets (the STAR benchmark). We detail our methods for filtering and processing data from the STAR benchmark in the "Dataset Curation" part of Section 3, and provide visualization samples of keyframes in Appendix A. The core methodology of our research, including the specific settings, intervention methods, and evaluation metrics for the three analytical circuits (Visual Information Auditing, Semantic Tracing, and Attention Flow), is thoroughly elaborated in Section 3 (Subsections 3.1, 3.2, and 3.3), which includes key mathematical formulas and parameter definitions. The theoretical framework supporting our experimental design is fully formalized in Appendix B. Furthermore, an anonymous interactive demo website is provided in Appendix D for reviewers to explore additional qualitative results. We believe these detailed descriptions are sufficient to support the reproduction of this work. All code will be made available upon acceptance of the manuscript.

| Token/Layer | Layer 1 | Layer 2 | Layer 3 | Layer 4 | Layer 5 |
|---|---|---|---|---|---|
|  | Institution | Архив | sierp | sierp | sierp |
| US | ona | пута | pert | tap | pure |
| ER | Session | пута | Session | 庄 | Ori |
| : | Portail | пута |  |  | Sav |
|  | ctl | пута | Carter | � | IR |
| <IMG001> | мп | мп | мп | мп | aks |
| <IMG002> | amen | amen | amen | alberga | amen |
| <IMG003> | ката | ката | ката | ката | ката |
| <IMG004> | jes | jes | jes | jes | jes |
| <IMG005> | anno | мп | мп | мп | anno |
| <IMG006> | 隆 | 隆 | 隆 | Sito | 隆 |
| <IMG007> | gresql | gresql | gresql | gresql | gresql |
| <IMG008> | ummy | ummy | ummy | ummy | ummy |
| <IMG009> | osz | osz | osz | Sito | anim |
| <IMG010> | opf | opf | opf | opf | opf |
| <IMG011> | pur | pur | pur | pur | pur |
| <IMG012> | ode | ode | ode | ode | ode |
| <IMG013> | wheel | wheel | wheel | ava | 置 |
| <IMG014> | arab | arab | ode | ode | arab |
| <IMG015> | gresql | gresql | gresql | end | ł |
| <IMG016> | кал | кал | CURLOPT | CURLOPT | cub |
| <IMG017> | way | way | way | way | way |

**Prompt**
USER: The input consists of a sequence of key frames from a video. Question: Which object was picked up by the person?
ASSISTANT: The object is the

**Answer:** Box

**Response:** Box of shoes

Figure 11: Qualitative example demonstrating the model's ability to recognize an object being picked up. The model correctly identifies the object as a "Box of shoes". The semantic tracing table displays the evolution of token representations across five layers that contribute to this accurate identification.

