# OpenReview forum: "CircuitProbe: Dissecting Spatiotemporal Visual Semantics with Circuit Tracing"
_ICLR.cc/2026/Conference — ICLR 2026 Conference Withdrawn Submission_

### Official Review · Reviewer_RRKb · 2025-10-25

**Soundness:** 2
**Presentation:** 1
**Contribution:** 2
**Rating:** 2
**Confidence:** 4

**Summary:**

This paper performs an analysis of how large visual language models (LVLMs) answer questions, with a focus on video data (vs. images).

There are 3 main takeaways:
1) Information about objects is represented in tokens corresponding to those objects’ locations.
2) Representations l_i corresponding to object tokens o_i at the i-th index become much more likely to be decoded as that object using the logit lens in the mid-to-late layers (see Figure 4).
3) LVLMs focus on context (ie tokens not corresponding to objects’ locations) in earlier layers and fine-grained details in middle layers when answering questions about visual dynamics.

Notably, takeaways 1 and 2 are identical to findings from “Towards Interpreting Visual Information Processing in Vision-Language Models” - Neo et al., ICLR 2025, which is not referenced in this submission.  Neo only considered image, not video, but it’s unclear if this is relevant, and I believe the experimental approach used here is also essentially the same.

**Strengths:**

I'm not sure what the current status is for interpreting video language models, but this work seems to make some contributions in this area (although they seem somewhat incremental given the overlap with Neo et al.).

I cannot understand Section 3.3 (which would be the most novel, I think), due to issues of presentation.

**Weaknesses:**

(major) Lack of novelty and poor treatment of related work:
- As mentioned, takeaways 1 and 2 are identical to findings from “Towards Interpreting Visual Information Processing in Vision-Language Models” - Neo et al., ICLR 2025, which is not referenced in this submission.  I believe the experimental approach used here is also essentially the same.
- Section 4.1 does almost nothing to connect the related works to this work.  It also includes the inaccurate statement: “Multimodal interpretability, particularly the analysis of visual-textual integration remains an under-explored frontier,” which is also in direct conflict with the first sentence of the abstract: “The processing mechanisms underlying language and image understanding in large visionlanguage models (LVLMs) have been extensively studied.”
- Section 4.2 has a similar problem.  I'm left unclear on what exactly the novelty of this work is meant to be and how it is related to previous works referenced.


(major) The paper is lacking critical details and experiments are not well explained or presented.
- “(3) Object Tokens: We select a set of tokens EO corresponding to those image patches fully contained within the object’s bounding box.” It is not described where the object's bounding box comes from.
- Section 2.1 paragraph on Video Pre-processing only describes how a frame is processed, not how a whole video is.  In general, the temporal aspect of the experiments is not well-described; it seems incidental to Section 3.1/takeaway 1, and I’m not convinced by the example in Figure 5 (section 3.2).  Section 3.3 seems most focused on video (and where most novelty would lie), but I found it extremely unclear.
- Section 3.3 states “This analysis is conducted under two distinct masking scenarios” (Object-centric masking and Contextual masking), but then talks about masking “fine-grained details”, which is not introduced or described.  Also, it is not stated what the task is.
- The background and experiment set-up is not well organized.  Information about what data is used is in the notation section.  Information about which models are used is in the introduction.  Both would be better placed at the outset of the experiments section.

(major) Conclusions are not well-supported:
- “Second, the symbolic text label provides a cleaner, more powerful semantic signal than the corresponding visual tokens. This confirms that the model’s performance is fundamentally dependent on object-level conceptual information.”
This seems incorrect, as the performance with the text token is lower than with the original visual object token (82.9% vs. 100%), if I understand correctly.  I think it would also be useful to consider an experiment where you insert the text token in a random location instead of in place of the masked visual object token.
- “Observation 5. LVLMs exhibits strong capabilities in capturing temporal dynamics for action recognition.” This example doesn't show that… Sitting and standing could just be static attributes of those frames.
- “These findings offer a coherent explanation of LVLM reasoning, moving beyond blackbox evaluations toward a more principled understanding.” This is overstated and also minimizes the contributions of prior works.

(moderate) Incorrect use of terms:
- 43: The limitations of these methods → The limitations of these works
- The use of the term "circuit" seems non-standard. It seems like it's being used to just refer to layers of the model, rather than subsets of neurons that perform particular computations.


(minor) Some sentences are very unclear/confusing:
- “We propose the hypothesis-driven methodology to dissect the internal computational circuits of LVLMs.” - unclear.  What is “the hypothesis-driven methodology”
- “This mask is applied to hypothesize and restrict attention between different token and layer subsets.” unclear.

**Questions:**

The most helpful thing would be to address the issues I mentioned above in the weaknesses.

---

### Official Review · Reviewer_13GA · 2025-10-29

**Soundness:** 2
**Presentation:** 2
**Contribution:** 1
**Rating:** 2
**Confidence:** 3

**Summary:**

This is an interpretability paper that seeks to probe how large vision-language models (LVLMs) perform spatiotemporal understanding.  In the first part, the authors mask out object tokens and evaluate the effect on final object label prediction.  In the second part the authors project intermediate states to visualize the vocabulary distribution it encodes.  In the last part, the authors perform object-centric and contextual masking across segments of the LLM model.

**Strengths:**

The strength of this work lies in its thorough interpretability exploration of the LLaVA model.

**Weaknesses:**

Firstly, the paper seems strangely formatted; the authors seem to have squashed the content of each page from the standard 53 lines per page to 46 lines per page).  On the surface this makes the paper appear to fill up the entire content length of 9 pages; but this reviewer would rather the authors be up-front about their content length with a smaller page count rather than manipulate formatting for appearances.  On the other hand, this reviewer is not sure if this breaks formatting rules for the ICLR conference.

The authors state that their goal is to understand mechanisms for spatiotemporal understanding in LVLMs (large vision-language models), but appear to only perform experiments on one particular choice - LLaVa.  It is unclear if these insights apply generally to arbitrary LVLMs as the authors seek as their scope, or just to this specific choice of LVLM.

Quite a few of the findings are quite unsurprising.  Firstly, the choice of image encoder as a CLIP ViT model inherently means that through the pretraining, the image representations are encouraged to have text-alignment.  Then, replacing them with explicit text embeddings in Circuit 1 seems like it should naturally help performance.  The authors should ablate over different choices of the image encoder, particularly when using image-only-pretrained image encoders such as DINO [1], SimCLR [2], etc. In particular, it has been shown that visual pretraining alone can help create representations that facilitate spatiotemporal reasoning [3] - motivated by this finding it seems appropriate to try vision-only pretrained image encoders.

Circuit 1 is naturally structured as an object recognition problem; the text prompt used is explicitly querying a subsequent object class name ("The object is").  It is quite unsurprising (Observation 1, Line 174) that when the object tokens are masked out the model struggles to answer correctly.  Naturally we expect such object-centric information to be encoded inside such object tokens.  Observation 2, as mentioned above, should be ablated with different image embeddings, as CLIP naturally leaks textual information - it is unsurprising that replacing it with corresponding text embeddings of the target object label allows the model to answer with the target object label.

Circuit 2 appears to just be an application of the logit lens approach (Nostalgebraist, 2020) from prior work.  The finding that abstract representations with spatial correspondence for image processing seems in line with findings of many old works in computer vision - the reviewer is not sure what is supposed to be considered novel here.

[1] Caron et al., Emerging Properties in Self-Supervised Vision Transformers, ICCV 2021.

[2] Ting et al., A Simple Framework for Contrastive Learning of Visual Representations, ICML 2020.

[3] Sun et al., Does visual pretraining help end-to-end reasoning?, NeurIPS 2023.

**Questions:**

The entire work appears to be quite manual; there needs to be manual effort in determining the object patches and then masking them out.  This makes the probing quite unscalable; are there ways to make these circuitprobing methods more automated?

---

### Official Review · Reviewer_aGNK · 2025-10-29

**Soundness:** 2
**Presentation:** 3
**Contribution:** 2
**Rating:** 2
**Confidence:** 2

**Summary:**

The paper presents CircuitProbe, a framework that analyzes how vision-language models understand space and time through three circuits: visual auditing (where object information localize), semantic tracing (how object semantic forms across layers), and attention flow (how context and details interact). The authors show that object tokens carry object identity, semantic meaning emerges mid-late layer, and reasoning follows a “context-first, detail-later” pattern.

**Strengths:**

1. The three-circuit structure (visual auditing, semantic tracing, attention flow) gives a coherent lens for analyzing how LVLMs process object concept within images and videos.

2. Few works have dissected *video-based* vision-language models at this mechanistic level. Applying circuit analysis to spatiotemporal reasoning is useful.

3. The paper is clearly written, well-illustrated the experiments support its findings.

**Weaknesses:**

1. **Limited novelty in findings.** The first two circuits mostly known properties from previous works [1, 2] , the object tokens contain semantic identity, and semantics become text-aligned in deeper layers.
2. **Limited evaluation scope.** Only LLaVA-based models and the STAR dataset are tested; it is unclear whether the results generalize to other LVLM architectures or datasets.
3. **No downstream utility.** Although the authors show that replacing visual tokens with their corresponding text tokens can slightly improve performance, the work mainly focuses on analysis without demonstrating how the circuit insights could improve model performance, reduce hallucinations, or inform model design.

---

[1] Neo, C., Ong, L., Torr, P., Geva, M., Krueger, D., & Barez, F. (2025). *Towards interpreting visual information processing in vision-language models*. In *Proceedings of the 13th International Conference on Learning Representations (ICLR 2025)*.

[2] Jiang, N., Kachinthaya, A., Petryk, S., & Gandelsman, Y. (2025). *Interpreting and editing vision-language representations to mitigate hallucinations*. In *Proceedings of the 13th International Conference on Learning Representations* (ICLR 2025).

**Questions:**

1. Do the same trends appear in other LVLM families (e.g., Qwen-VL, VideoChatGPT)?
2. Could the authors connect their metrics (CR, AP) to downstream task performance to demonstrate practical interpretability benefits?

---

### Official Review · Reviewer_RrQL · 2025-11-07

**Soundness:** 3
**Presentation:** 3
**Contribution:** 1
**Rating:** 2
**Confidence:** 4

**Summary:**

The work investigates internal representations and mechanisms of LLMs trained for video understanding. Specifically, the paper explores the grounding of object and action information in vision tokens, their grounding to semantic concepts, their evolution across the model’s hierarchy, and the reliance of their emergence in attention mechanisms.

**Strengths:**

The paper is clearly written. The experiments carried out are conceptually solid and rigorous.

**Weaknesses:**

- The novelty of this work is limited, taking into account previous work that was missing from the references in the manuscript. In particular, the work of “Towards Interpreting Visual Information Processing in Vision-Language Models” (ICLR, 2025) and "Interpreting and Editing Vision‑Language Representations to Mitigate Hallucinations” (ICLR, 2025) uses similar approaches for analyzing questions posed by this paper.

- The submission only explores a limited number of models from the same family (LlaVA).

- The paper only explores one dataset (STAR benchmark). Results may not generalize to datasets involving other objects, scenarios, visual properties, etc.

- The framing of spatiotemporal understanding seems misleading. The work seems to be investigating the grounding of objects and actions in “vision” tokens (which the above mentioned papers also do). It does not explore the encoding of spatial relationships (e.g. object layouts, relative positions), temporal sequences, or others.

**Questions:**

- Clarification on what insights/approaches are different from the following work: “Towards Interpreting Visual Information Processing in Vision-Language Models” (ICLR, 2025) and "Interpreting and Editing Vision‑Language Representations to Mitigate Hallucinations” (ICLR, 2025) should be included. These works have already performed detailed analyses of internal visual token representations, localization of object information, and latent‐space editing in image‐based VLMs. Merely extending these approaches to video modalities seems trivial given the same backbones are being utilized in both types of work. Specially if the work does not handle temporal encoding, cross‐frame object trajectories, or motion cues. Also, these papers should be included in the discussion of previous work.

- Why not extend these approaches to analyze other model architectures and variants?

- How do the findings generalize  to domains beyond the STAR benchmark scenario?

Minor comments:
- Typo in Figure 2: Object Tokns

---

### Note · Authors · 2025-11-15

I have read and agree with the venue's withdrawal policy on behalf of myself and my co-authors.